# Molecular Evolution, Neurodevelopmental Roles and Clinical Significance of HECT-Type UBE3 E3 Ubiquitin Ligases

**DOI:** 10.3390/cells9112455

**Published:** 2020-11-10

**Authors:** Mateusz C. Ambrozkiewicz, Katherine J. Cuthill, Dermot Harnett, Hiroshi Kawabe, Victor Tarabykin

**Affiliations:** 1Institute of Cell Biology and Neurobiology, Charité-Universitätsmedizin Berlin, Corporate Member of Freie Universität Berlin, Humboldt-Universität zu Berlin, and Berlin Institute of Health, Charitéplatz 1, 10117 Berlin, Germany; Katherine.cuthill@charite.de; 2The Integrative Research Institute for the Life Sciences, Institute of Biology, Humboldt-Universität zu Berlin, 10115 Berlin, Germany; Dermot.Harnett@mdc-berlin.de; 3Department of Pharmacology, Gunma University Graduate School of Medicine, Maebashi, Gunma 371-8511, Japan; Kawabe@em.mpg.de; 4Institute of Neuroscience, Lobachevsky University of Nizhny Novgorod, pr. Gagarina 24, 603950 Nizhny Novgorod, Russia

**Keywords:** ubiquitin, E3 ubiquitin ligase, UBE3A, UBE3B, UBE3C, Angelman syndrome, Kaufman oculocerebrofacial syndrome, autism spectrum disorder

## Abstract

Protein ubiquitination belongs to the best characterized pathways of protein degradation in the cell; however, our current knowledge on its physiological consequences is just the tip of an iceberg. The divergence of enzymatic executors of ubiquitination led to some 600–700 E3 ubiquitin ligases embedded in the human genome. Notably, mutations in around 13% of these genes are causative of severe neurological diseases. Despite this, molecular and cellular context of ubiquitination remains poorly characterized, especially in the developing brain. In this review article, we summarize recent findings on brain-expressed HECT-type E3 UBE3 ligases and their murine orthologues, comprising Angelman syndrome UBE3A, Kaufman oculocerebrofacial syndrome UBE3B and autism spectrum disorder-associated UBE3C. We summarize evolutionary emergence of three *UBE3* genes, the biochemistry of UBE3 enzymes, their biology and clinical relevance in brain disorders. Particularly, we highlight that uninterrupted action of UBE3 ligases is a sine qua non for cortical circuit assembly and higher cognitive functions of the neocortex.

## 1. Development of the Neocortical Circuits

Uninterrupted communication between executive brain centers, on a molecular and single neuron level manifested as circuit activity, allows for integration of external and internal sensory stimuli and generation of appropriate efferent responses. The neocortex, paramount achievement of mammalian brain evolution, constitutes a biological substrate for higher cognitive abilities in *Primata*. Spatiotemporally synchronized patterns of activity in and between neuronal circuits are enabled by tightly controlled transcriptional–translational and morphoregulatory programs in developing neocortex [1].

Assembly of a neuronal circuits in the mammalian neocortex starts during early neurogenesis, where at the ventricular zone (VZ) of the neural tube, radial glial cells (RGCs), give rise to postmitotic cells that differentiate into glutamatergic excitatory neurons. One of the most prominent features of cortical neuron development is that the immature nerve cells born close to the VZ do not stay in situ but migrate radially to distribute horizontally into their target niches to establish neocortical layers. These cortical niches contain neurons originating from spatiotemporally defined RGC populations. Neurons of each layer share transcriptional signatures and display similar physiological properties and patterns of connectivity. Molecular signaling orchestrating the postmitotic differentiation is conserved between neurons of different layers. The initial transcriptional ground states in RGCs, often display very subtle differences, but represent spatiotemporally restricted developmental trajectories that seem to be the source of the excitatory neuronal subtype variations in the cerebral cortex [2,3]. Formation of cortical layers, which requires orthodox supervision of neurogenesis, neuronal migration, specification of a single axon and its guidance, formation and elaborate growth of dendritic arbor and synaptogenesis, is therefore fundamental for establishment of afferent inputs and efferent connectivity of the cortical circuits. Specification of different neuronal types and the intricate geometry of the neuron dictates its contribution to the activity of the neuronal circuit [4]. Assembled circuits in and outside of the cortex underlie higher cognitive functions. A well characterized physiological manifestation of such is long-term potentiation (LTP), the most studied form of synaptic plasticity associated with memory storage, that leads to long-term strengthening of synaptic transmission between neurons driven by a specific pattern of activity [5]. Formation and retrieval of memories represents another aspect of circuit activity, as it requires directional coupling between two brain regions, neocortex and hippocampus. On a circuit physiology level, these processes are thought to be supported by neocortical desynchronization of alpha/beta rhythm and synchronization of hippocampal theta/gamma oscillations [6,7].

## 2. Neuronal Circuit Assembly and Neurodevelopmental Disorders

Aberrant assembly of neuronal circuits represents the primary cause of higher cognitive dysfunction [4]. Malfunctioning of neuronal circuits associated with structural changes in neuronal morphology is a physiological underpinning of neurodevelopmental disorders. These include autism spectrum disorders (ASD), or intellectual disability (ID), both extremely heterogenous and multifactorial groups of diseases, with their psychiatric and cognitive comorbidities and ever-increasing prevalence [4]. Genetic predisposition combined with an environmental impact are the core ASD/ID etiology [8].

## 3. Protein Ubiquitination as a Potent Regulatory Principle

Neuronal transcriptional signatures are the effect of a discrete regulation of gene expression. Recent reports derive the sources of cortical neuron complexity using state-of-the-art single cell transcriptomics to cluster the types of cellular populations present in the developing brain. These elegant but somewhat reductive approaches turn a blind eye to the involvement of post-transcriptional modes of gene expression regulation in the brain development. Recent accumulating evidence reveals critical roles of post-translational modification of ubiquitination in the formation of neuronal networks [9,10].

## 4. Protein Ubiquitination Is a Reversible Post-Translational Modification

Since its discovery in 1975 [11], ubiquitin (Ub) has been linked to regulation of the broad range of cellular processes, from DNA repair to intracellular trafficking. A breakthrough discovery that protein ubiquitination results in 26S proteasomal targeting and substrate degradation was awarded a Nobel Prize in Chemistry in 2004 and shed light onto the complex biochemistry of the modification [12]. Ubiquitination involves conjugation of 8.6 kDa globular Ub to a substrate protein in a cascade of enzymatic reactions catalyzed by an E1 Ub activating enzyme, an E2 Ub conjugating enzyme, and an E3 Ub ligase. Of note, apart from the canonical successive sequential mode of Ub shuttle, additional biochemical scenarios of ubiquitination have emerged plausible as well [13]. Eventually, Ub is shuttled onto the substrate protein, a reaction catalyzed by the E3 ligases, which determine the specificity of the ubiquitome [14]. Canonically, Ub is linked to the substrate via an isopeptide bond between C-terminus glycine residue of Ub and a lysine (K) of the target protein. Non-canonical ubiquitination includes linkages employing the N-terminal methionine, serine, or threonine of the substrate protein [15]. Potential functional consequences of ubiquitination via the non-canonical sites remain yet to be described. K residues on substrate proteins can be conjugated with a single Ub or with polyUb chains (polyubiquitination). Homotypic polyUb chain can be formed through one of the seven K residues of Ub (K6, K11, K27, K29, K33, K48 and K63). The type of the polyUb chain specifies the functional consequence of ubiquitination. In the mouse brain, approximately 60% of Ub remains a free monomer, 35% is detected as a monoUb on substrates and 5% in polyUb chains [16].

Protein ubiquitylation is reversible and catalyzed by ubiquitin proteases, enzymes called deubiquitylases (DUBs). Human genome harbors some 90 loci encoding for DUBs, far less than for E3 ligases, indicating a rather broad biochemical promiscuity. However, some DUBs display a strong ubiquitin chain type affinity, i.e., they catalyze proteolytic cleavage of certain polyUb chains, rather than the ubiquitylated substrate per se [17].

## 5. Uninterrupted Function of E3 Ligases Is Fundamental for Brain Development

Given a distinct mode of catalysis, E3 ligases fall into two, or three main classes. Homologous to E6-AP (HECT) type E3 Ub ligases act as direct Ub acceptors, conjugating it to their C-terminal cysteine residue, before shuttling the Ub onto their substrates. HECT is a group of biochemically distinct enzymes with 28 members in humans identified to date. Our previous works signify the involvement of HECT-type E3 ligases in the most fundamental neurodevelopmental processes including establishment of dendritic arbor [18], neurite extension [19], and acquisition of axon and neuronal migration [20]. Of all E3 ligase genes, approximately 13% are mutated in a neurological disorder, with ASD/ID-associated mutations in E3 ligase genes representing the biggest group [21]. Among these are the brain expressed UBE3 enzymes: the founder of the subclass Ubiquitin Ligase E3A (UBE3A), with its gene loss-of-function resulting in Angelman syndrome (AS) and Ubiquitin Ligase E3B (UBE3B), linked to Kaufman oculocerebrofacial syndrome (KOS) as well as ASD-associated Ubiquitin Ligase E3C (UBE3C). In this review, we will focus on highlighting their association with severe neurodevelopmental disorders and their function in the assembly of neuronal circuits. We will discuss the fact that despite a high degree of homology at the level of peptide sequence, there is strikingly little functional redundancy between these three ligases. Our previous review covers the neuronal roles of UBE3 ligases [9]; we will therefore focus on the most recent published works here.

## 6. UBE3 Ligases Are Able to Assemble Different polyUb Chain Types

Classically, ubiquitination of a substrate protein has been linked to its degradation. Loss-of-function mutations in E3 ligases are associated with neurological diseases, implicating the pivotal role of specific degradation in fundamental neurodevelopmental pathways ergo cortical circuit assembly. Recent data sheds lights onto non-degradative ubiquitination, encoded by polyUb chain type conjugated to the substrate, indicative of these types of modifications at play during neuronal development. The K48- and K63-linked polyUb chains constitute the major types of polyUb chains in the mouse brain and human frontal cortex and K11-linked chains are three- to four-fold less abundant [16]. Canonical ubiquitination, involving conjugation of K48-linked Ub chain destines substrate proteins for proteasomal degradation, whereas functional consequences of K63-linked chain conjugation include non-proteasomal effects, such as trafficking of transmembrane proteins, lysosomal degradation, or in their mono form, modification of histones to organize chromatin structure [10,22]. Although there is sufficient evidence for physiological consequences of canonical K48- and K63-linked ubiquitylation, roles of other polyUb chain types remain elusive, especially in terms of the development of the nervous system. Of note, K6-, K27-, K29-, and K33-polyUb chains together represent less than 1% of total Ub found in the mouse brains and human frontal cortex [16].

Whereas monoUb has been linked to modulating enzymatic activity or protein–protein interactions [18], different polyUb linkages display fundamentally opposite biochemical conformations, which may explain the various substrate protein fates that they predestine towards [23]. Structurally, molecular interfaces between hydrophobic patches on adjacent Ubs in a dimer can be formed by K6, K11, K29, K33, and K48, generating a conformational landscape for interactions. Predominantly, K63-linked chains adopt an open conformation with two Ub molecules only contacting at the linkage site [24].

The determinants of chain type formed by a given E3 Ub ligase have been an area of debate. Regarding the HECT ligases, these are the function of the catalytic HECT domain and do not seem to depend on the E2 conjugating enzyme, but rather are located within the 60 amino acids of the C-terminus of the HECT domain C-lobe [25]. Within the HECT family, UBE3A preferentially forms K48 polyUb chain [25], Ube3b is able to form both K48- and K63-linked chain [26] and UBE3C catalyzes K48- and less preponderant K29- and K11-linked chains [27]. K29 chains exist in cells within heterotypic and branched polyUb chains, among other Ub species [28]. Such divergent biochemical properties of these homologs are somewhat surprising, given a high degree of amino acid sequence similarity between their HECT domains. Indispensable roles of each UBE3 ligase in a neuron are associated with inherently specific biochemical modes of action (Figure 1).

## 7. Functional and Pathophysiological Implications of the HECT Domain Organization of UBE3 Ligases

Understanding of neuronal roles of UBE3s requires an in-depth analysis of the biochemistry of their action. Transient binding of Ub (autoubiquitination) prior to its sequential transfer onto the substrates, is a distinct feature of HECT-type ligases. Evolutionarily conserved organization of the catalytic C-terminal HECT domain highlights its E2 conjugating enzyme-interacting N-lobe, a short flexible linker and a C-lobe, containing an Ub acceptor cysteine [29]. Conformational state of the HECT domain determines Ub binding and allows for proper orientation of the growing polyUb chain [30]. Unsurprisingly, amino acid substitutions in this domain engender dramatic effects for the enzymatic activity of a given E3.

We know a fair amount regarding the organization of UBE3A HECT domain and its mode of action [31]. Biochemical studies show a two-step mechanism, with a site for initial E2 binding and another for Ub chain elongation [32]. Moreover, fully catalytically active UBE3A is a trimer with the oligomerization interface using a HECT domain localized residue. Destabilization of the trimer formation also seems to be a feature of a specific loss-of-function Angelman syndrome mutation [33]. An AS-associated transversion mutation found in the HECT domain of UBE3A I827K destabilizes protein folding and causes C-lobe aggregation, which hinders UBE3A enzymatic activity [34]. A specific c-Abl-mediated tyrosine residue phosphorylation in UBE3A HECT domain has also been linked to its oligomerization and activity [35]. Until now, no indications for oligomerization of UBE3B or UBE3C exist.

KOS-associated point mutations in *UBE3B* leading to single amino acid substitutions in the HECT domain appear to affect the enzymatic activity of the ligase implicating conformational restrictions. Theoretical modeling of Q727P substitution predicts it renders the HECT domain unfit for substrate binding and positioning towards catalytic cysteine, required for efficient ubiquitination [36]. HECT-localized KOS-linked substitutions G779R and R997P fail to restore deficient dendritic arbors of *Ube3b* conditional forebrain-specific knockout (cKO) neurons, and the latter one exhibits altered subneuronal localization, when compared to the native UBE3B [26]. It is possible, that perinuclear localization of R997P UBE3B represents its ER-trapped folding-deficient form, similar to the case of another postsynapse-associated ASD-linked protein, neuroligin-4 [37]. The scenario of ER-localized processing of UBE3B should be verified experimentally.

Recent studies have shed light on the crystal structure of UBE3C [38]. Aside from the catalytic cysteine, an additional K903 site is also a major site of autoubiquitination in its HECT domain, whereas Q961, S1049, and three C-terminal amino acids are necessary for UBE3C activity [38]. Additionally, the N-terminal region just preceding the HECT domain is essential for stabilization and activity of UBE3C. Of note, ASD-associated point substitutions in UBE3C localize to its HECT domain. It is tempting to speculate the conformational bias of these mutants and inefficient UBE3C-mediated ubiquitination (Table 1).

## 8. Developmental Expression and Molecular Evolution of UBE3 Ligases

There has been a slight confusion about the functional homology between UBE3s, as observed in introductions to multiple publications. All three UBE3s share a gross domain organization plan including the catalytic HECT domain and long stretches of low complexity, possibly accounting for the high interactivity of ligases with binding partners [41]. UBE3B and UBE3C display an N-terminal IQ motif. Apart from this, evolutionary, biochemical, and physiological comparison of these three ligases indicates that each of them represents a distinct biochemical entity with specific functions in a neuron.

Single cell RNA sequencing databases provide insights into spatiotemporal expression dynamics of *Ube3* genes. These expression trajectories indicate that each mRNA is present in progenitors and postmitotic neurons in the developing murine neocortex [3]. Ube3a mRNA shows elevated levels in the latter cell type, Ube3b mRNA exhibits a slight decrease in its levels in postmitotic cells and Ube3c mRNA displays an enrichment in E12 multipotent progenitors, able to give rise to neurons of both upper and deeper layers of the neocortex (Figure 2). From the evolutionary biology point of view, UBE3A is more closely related to members of the small HERC family of ligases than to UBE3B and UBE3C [42].

Phylogenetic analyses show that HECT E3s appear before animals or very early in their evolution and appearance of them precede the emergence of the cellular function that they control [43]. That said, evolutionary emergence of *Ube3*s dates long before the origin of the central nervous system. The history of *Ube3a* seems more tumultuous than that of *Ube3b* and *Ube3c*. Orthologues of *Ube3a* are identified in bilaterians, including vertebrates and fruit fly, but not in *C. elegans* indicating *Ube3a* loss in some nematodes [43]. Additionally, *Ube3a* shows secondary loss in choanoflagellate lineage. On top of this, the origin of *Ube3a* imprinting seems to be acquired in the mammalian radiation. *Ube3b* appeared relatively early in evolution and is present in fungi and other opisthokonta. Of the three *Ube3*s, *Ube3c* seems to be the youngest with its orthologues found in *Bilateria* and other *Metazoa* [44] (Figure 3). The phylogenetic searches for the ancestors of postsynaptic proteins demonstrate the presence of core protein orthologues in simple metazoans and choanoflagellates devoid of or with a primitive nervous system. This indicates an ancient origin of genes, including *Ube3*s and their orthologues, implicated in human neurodevelopmental disorders, which seem to regulate key cellular housekeeping processes in primitive organisms [42]. This might explain the critical role of UBE3s outside of the nervous system, as well.

## 9. The Founding Member of HECT Family, E6AP/UBE3A, Angelman Syndrome, and Its Role in the Developing Nervous System

The gene encoding for the founding member of HECT type ligase family, E6-associated protein, *E6AP*, or *UBE3A*, is imprinted in neurons in mice and humans resulting in maternal expression of *UBE3A* allele. Inactivation, deletions or mutation of maternal *UBE3A* allele are causative of Angelman syndrome (AS) with, among other symptoms, severe intellectual disability, speech impairment, behavioral disturbances, and developmental delay [45]. Duplications of 15q11.2–q13.3 locus, including *UBE3A*, is a genetic cause of the ASD [46]. We summarized some of the less recent findings on neuronal role of *UBE3A* in our previous review [9]. In human cortex, UBE3A is expressed in glutamatergic and GABAergic neurons and to the lower extent in glia [47]. Maternal *Ube3a* KO in mice (*Ube3a*
^m−/p+^) recapitulates features of AS patients. Loss of *Ube3a* leads to ineffective LTP in the hippocampus, indicative of defects in neuronal circuitry associated with learning and memory [48].

## 10. Mother Allele-Specific Neuronal Expression of *UBE3A*

While *UBE3A* is located on chromosome 15, the mode of AS inheritance is not typical autosomal dominant or recessive [49,50]. Paternal alleles of human *UBE3A* and mouse *Ube3a* are silenced by large non-cording antisense RNA, expressed from the nearby loci [49]. The murine non-coding RNA called Ube3a antisense transcript (*Ube3a*-ATS) is transcribed by an atypical RNA polymerase type II and covers the entire *Ube3a* locus. Deletion of maternal *Ube3a* causes the complete lack of Ube3a protein expression, restored by additional targeting of paternal *Ube3a*-ATS [51]. *Ube3a*-ATS is a critical suppressor of the *Ube3a* expression from the paternal allele, which is likely the case in humans as well. Interestingly, *Ube3a*-ATS is expressed solely in neurons, without affecting expression of Ube3a in glial cells [52,53]. Of note, accumulating evidence suggests that the disinhibition of paternal allele of Ube3a expression offers a potential therapeutic approach for AS [54,55].

## 11. Molecular Biology of UBE3A, a Proteasome Regulator and Modulator of Cellular Signaling

Other reviews [56] highlight the mechanistic function of UBE3A, describing its association with several proteasome receptors, like DDI1 [57,58] or PSMD4 but also with proteasome itself, which is thought to control the proteolysis [59]. AS-associated point mutations in UBE3A N-terminus show impaired PSMD4 binding [60]. Further, AS-linked UBE3A^T485A^ reduces proteasome activity and leads to a stabilization of β-catenin, indicative of its enhanced ability activate Wnt signaling [61]. The T485 residue is phosphorylated by protein kinase A (PKA) to (auto)-inhibit UBE3A, so that T485A engenders enhanced turnover of substrates in patient-derived cells and intensifies development of dendritic spines [62]. UBE3A has also been found to ubiquitinate proteasome related protein Rpn10, implicating UBE3A in a robust regulation of proteostasis [63]. As is the case for UBE3B, UBE3A also seems to play pivotal role in maintenance of mitochondrial function [64]. Quite intriguing is association of UBE3A with another HECT E3, HERC2 and formation of a high molecular weight complex of yet not understood function [65]. Upon interaction, HERC2 stimulates UBE3A enzymatic activity in a ubiquitination-independent manner [66]. There is a significant body of work on UBE3A in immune response to viral infection, cancer, and neurodegeneration. Unfortunately, we cannot mention all published works on UBE3A here, solely due to space limitations. Further on, we will focus on the roles of *UBE3A* regarding the assembly of cortical circuits for the sake of its functional comparison with *UBE3B* and *UBE3C*.

## 12. Isoform-Specific Neuromorphoregulatory Roles of UBE3A

Isoform- and tissue context-specific morphoregulatory roles of UBE3A have been fundamental for our understanding of its contribution to the development and functions of neuronal circuits. Five isoforms of UBE3A mRNAs were identified with three of the transcripts coding for protein isoforms [67]. *Ube3a* downregulation leads to less complex dendritic arbors in the cortex and hippocampus and abrogates growth of apical dendrites associated with loss of polarization of Golgi apparatus. Such phenotypes are reversible by re-expression of isoform 2, but not of isoform 1 or 3 of Ube3a [68]. In contrast to AS neurons, UBE3A-overexpressing neurons do not exhibit polarization problems [69]. Another report provides evidence that isoform 1 accounts for the majority of UBE3A in neurons [70]. Moreover, knockdown of alternatively spliced Ube3a1 devoid of its catalytic activity prompts robust dendritic growth, reduces the volume of spines and leads to decreased synaptic transmission. This function of Ube3a1 is encoded in its 3′ untranslated region, which acts as a competitive scavenger of microRNAs, specifically of miR-134 [71].

Expression of different isoforms of UBE3A has its relevance in the pathophysiology of AS. Developmental well-being of AS patients seem to rely on the isoforms of UBE3A affected in the genome. For instance, patients with Met1Thr mutation eliminating isoform 1 of maternal UBE3A exhibit far more advanced developmental skills as compared to other non-mosaic AS patients [72].

## 13. UBE3A Acts Locally at the Spine and Globally in the Nucleus to Regulate Synaptic Transmission

Ube3a localizes at the axon terminals and at the head and neck of dendritic spines [47] to regulate their density and length, glutamatergic and activity-dependent transmission [73]. Its ubiquitination targets degraded at the proteasome include Arc [48], responsible for endosomal recycling of α-amino-3-hydroxy-5-methyl-4-isoxazolepropionic acid receptors (AMPARs), and phosphorylated ephexin-5 [74], a negative regulator of excitatory synapse development. Recently, Protein Phosphatase 2 Phosphatase Activator (PTPA) has joined the group of bona fide UBE3A substrates. *Ube3a*
^m−/p+^ mice exhibit increased PTPA level and protein phosphatase 2A (PP2A) activity. Normalizing PTPA levels, as well as pharmacological inhibition of PP2A reversed developmental defects of excitatory synapses in AS mice [75]. Regarding other cellular pathways, Ube3a-mediated p18 targeting negatively regulates mTORC1 signaling and normalizing p18 overexpression in AS mouse model restores dendritic spine maturation, LTP and learning [76]. BMP repression is also involved in Ube3a-regulated synaptic growth in *Drosophila* [77].

Given *UBE3A* involvement in a developmental syndrome presenting a severe ID, a significant body of literature addresses synaptic roles of the ligase. Recently, extensive local synaptic actions of Small Ub-Like Modifier 1 (SUMO1), have been rebutted since in neurons, SUMO1 was found mainly in the nucleus [78]. Recent research on UBE3A also identifies its cytoplasmic and nuclear isoforms, with nuclear UBE3A as the predominant form, indicative of its nuclear roles, including regulation of transcription. Loss of UBE3A nuclear isoform, but not the cytoplasmic one, recapitulates behavioral and physiological phenotypes described for maternal *Ube3a* KO mouse [79]. Mimicking of the ASD-linked 15q11–13 chromosomal triplications by increasing nuclear UBE3A leads to diminished expression of excitatory synapse organizer cerebellin precursor 1 (Cbln1), associated with sociability in mice. Restoration of Cbln1 in midbrain ventral tegmental glutamatergic neurons alleviates *Ube3a* loss-linked sociability deficits [80]. It is possible that UBE3A acts to maintain proteostatic surveillance locally at single synapses but also globally regulate neuronal physiology by operating at the euchromatin-rich domains [56].

In line with this, UBE3A suppresses neuronal hyperexcitability by degradative ubiquitination of Ca^2+^- and voltage-dependent big potassium (BK) channels in human neurons and brain organoids. Inhibition of BK channels in AS mice alleviates the seizure susceptibility demonstrating the channelopathy as underlying cause of AS network dysfunction [81]. Moreover, UBE3A-mediated endocytosis of small conductance potassium channels (SKs), is critical for NMDAR activation ergo hippocampal LTP. Learning and memory deficits in AS mice are reversible by blocking SK2. This mechanism elucidates that SK channelopathy also contributes to cognitive aberrances and network dysfunction in AS [82].

## 14. Critical Developmental Window of UBE3A Expression

Neuron-restricted imprinting of UBE3A offers ways of intervention in AS mouse models, which have been useful to determine that pharmacological un-silencing of paternal allele in the adulthood is possible [54,55], yet proves ineffective in rehabilitating the cognitive abilities. Strikingly, rescuing anxiety, stereotypies and epilepsy in AS mice requires reinstatement of UBE3A during early development, but not juvenile or adult stages. This is in line with the findings that early embryonic deletion of *Ube3a*, but not its juvenile or adult loss, recapitulates all behavioral deficits of AS mice [83]. There seems to be no requirement of the developmental window for UBE3A reinstatement for the restoration of hippocampal LTP [84], or excitation/inhibition balance identified in mouse prefrontal cortex in AS mouse model [85]. GABAergic but not glutamatergic loss of *Ube3a* is responsible for perturbed excitation/inhibition imbalance with enhanced seizure susceptibility, signifying inhibitory neuron-specific loss of *Ube3a* as a principal reason for hyperactive neuronal circuits in AS mouse model [86].

## 15. Neuronal Phenotypes and Physiological Relevance of UBE3A Gain-of-Function

As mentioned before, maternal copy number gains of 15q11.2–q13.3 lead to Dup15q syndrome, characterized by intellectual disability, developmental delay, muscle hypotonia and speech impairment. Dup15q represents the most penetrant chromosomal anomaly in ASD patients accounting for 1–3% cases worldwide. UBE3A belongs to approximately 40 genes in that chromosomal region and its gain-of-function is thought to represent the molecular cause in Dup15q patients [87]. Indeed, transgenic mice, overexpressing isoform 2 of UBE3A in the glutamatergic neurons of the forebrain exhibited behavioral traits relevant to ASD [88]. Increased dosage of UBE3A in primary neuronal cultures or in the Ube3a ASD mouse model abrogates dendritic growth and synapse number in a mechanism involving caspase-3 activation and tubulin cleavage [89]. UBE3A has also been identified as a negative regulator of retinoic acid synthesis. Importantly, in animals with hyperactivation of UBE3A in the prefrontal cortex, retinoic acid supplements alleviated ASD-like phenotypes [90].

## 16. The Case of Forward Genetics: Kaufman Oculocerebrofacial Syndrome (KOS) and *UBE3B* Gene Discovery

In 1971, Robert Kaufman described four of seven siblings, who exhibited severe mental retardation with absent speech, characteristic facial dysmorphisms with blepharophimosis, aberrances of the eye, mongoloid slant of the palpebral fissures and small mandible. Already then, an autosomal recessive mode of disease inheritance seemed the most likely [91], later confirmed formally. Since then, cases of KOS, previously also known as blepharophimosis-ptosis-intellectual-disability syndrome, have been reported worldwide (OMIM #244450). Distinct features of KOS include face and eye malformations, microcephaly, severe psychomotor retardation, and growth arrest. Additionally, patients often display serum hypocholesterolemia, indicative of downregulation of cholesterol synthesis. Neurologically, hypoplastic or aplastic corpus callosum and anterior commissure, Chiari type I malformation, smaller pituitary gland, and seizures are found in some cases. KOS patients suffer from severe ID with absent speech [36,92].

Initially, two consecutive reports have revealed loss-of-function mutations in *UBE3B*, which inherited in autosomal recessive manner, are causative for KOS [36,93]. Since then, more KOS cases and novel mutations have been described, some of them resulting in frame shift and premature translation stop, but some of them leading to a single amino acid substitutions [92,94].

*UBE3B* gene was discovered as a gene upregulated after acoustic trauma in the chick basal papilla. Its expression dramatically increased in the lesion in chick inner ear, but not in undamaged regions [95]. Human *UBE3B* generates two splice isoforms, one devoid of the HECT domain, presumptively non-functional [96]. Notably, UBE3B has also been identified in a locus for a delayed-onset, progressive, high-frequency, nonsyndromic sensorineural hearing loss [97]. Some of the KOS patients also display hearing impairment [92]. Further research on UBE3B should address its possible molecular roles in the inner ear.

## 17. UBE3B in Non-KOS Diseases

Overall, UBE3B is highly expressed in the central nervous system, digestive tract, respiratory system, as well as in multiple cell lineages of skin and other soft tissues. Severe developmental delay in KOS patients and multi-organ defects signify the involvement of UBE3B in systemic homeostasis. Similarly to KOS, G>A polymorphism in cattle resulting in exon skipping and in-frame deletion of a portion of catalytic HECT domain is associated with ptosis, intellectual disability, growth retardation and increased mortality [98]. Besides KOS, UBE3B mutations have been identified is ASD patients [99]. Additionally, possible involvement in pathophysiology of schizophrenia onset further supports the pivotal role of UBE3B in formation of cortical circuits [100].

## 18. Molecular Context of UBE3B Actions in Neurons

Soon after *UBE3B* loss-of-function was discovered causative of KOS [36,93], the quest for unveiling its cellular and molecular roles, particularly in the central nervous system, began. Initial report on UBE3B explores its functions as an enzyme in the cell and proves its E3 ubiquitin ligase activity. Point mutation of the C-terminal catalytic cysteine or loss of the catalytic HECT domain leads to abrogation of the ubiquitin binding, indicative of the loss of enzymatic activity. As predicted from the domain organization, N-terminal IQ motif of UBE3B also interacts with calmodulin, and this interaction results in reduction of the ligase ubiquitination activity. Accordingly, deleting the IQ domain follows with an increase in the enzymatic activity of UBE3B, possibly in a mechanism involving a removal of self-inhibitory state due to the N-terminus sequestration of the HECT domain in a steady autoinhibited state, characteristic for some HECT-type ligases [101]. Contrary to the described role of the IQ domain as a calcium-independent calmodulin interaction site [102], variation of intracellular calcium seems to modulate calmodulin-UBE3B interaction, ergo UBE3B activity. How exactly calcium contributes to regulation of UBE3B function remains to be investigated. Regarding cellular physiology, knock-down (KD) of UBE3B in cells engenders mitotic arrest of cells and reduced cell survival associated aberrant punctate mitochondrial morphology and altered physiology [103].

The findings on UBE3B-mediated mitochondrial function and intracellular Ca^2+^ are especially relevant in the light of UBE3B implications in the stress response and cellular viability. In the first paper identifying mutations in *UBE3B* as a cause of KOS, researchers demonstrate that *C. elegans* ortholog of *UBE3B*, oxi-1 is a player in cellular protection under oxidative stress [36]. Moreover, siRNA-mediated UBE3B KD in human glioblastoma cells results in their sensitization to reactive oxygen species inducing agent, temozolomide [104]. Additionally, expression of IQ domain-devoid UBE3B, which displays higher enzymatic activity, in human cells induces apoptosis [103]. Given a reported higher enzymatic activity of this deletion mutant of UBE3B elsewhere, it would be worthwhile to test the link between its ubiquitination hyperactivity and cell death.

## 19. UBE3B as a Master Regulator of Metabolic Homeostasis

The next report on UBE3B function focuses on its role in the metabolic pathways and links it to the development of the central nervous system. In their previous report, this group of researchers first identified *UBE3B* as a candidate ASD gene in a whole-exome sequencing and homozygosity analysis of 16 probands. Additionally, Ube3b mRNA level increased in E16.5 mouse cortical neurons at DIV6 (day-in-vitro 6) after KCl-induced hyperpolarization, again linking UBE3B, neuronal excitability and Ca^2+^ [99]. Next, the authors follow up on the ligase and its role in metabolic homeostasis and nervous system using a full KO mouse model. Loss of *Ube3b* in mice leads to severe developmental defects including anatomical, behavioral, and metabolic disturbances, including growth retardation, hypoplasia of corpus callosum, enlarged ventricles and decreased cortical thickness. Using Golgi–Cox labeling, authors show that cortical neurons in *Ube3b*^−/−^ mice display reduced dendritic branching and spine density. Using a cell line stably expressing tagged UBE3B, affinity chromatography and mass spectrometry, authors identify branched-chain α-ketoacid dehydrogenase kinase (BCKDK) as a putative UBE3B target, further validated in ubiquitination assays. BCKDK level was also increased in *Ube3b*^−/−^ cortex, liver and skeletal muscle, indicative of a specific pathway for UBE3B-mediated BCKDK degradation. Consequently, authors demonstrate that metabolite profiles of *Ube3b*^−/−^ mice display signs of severe metabolic perturbations, with high levels of, among others, plasma spermidine, and S-adenosylhomocysteine, and cortical urea. Authors also show that patients bearing compound heterozygous mutations in *UBE3B* display similar metabolic perturbations, as identified in mice. Mitochondrial physiology, as demonstrated before in a cell line, is affected in skeletal muscle of *Ube3b*^−/−^, but not in the liver. That is particularly interesting, given some indications of UBE3B indispensability for skeletal muscle function [105]. Behavioral phenotypes in the *Ube3b*^−/−^ mice include loss of vocalizations, decreased grip strength and decreased grooming and nesting. The interpretation of behavioral changes is, however, not straightforward, as severe developmental and anatomical defects, likely account for poor sensorimotor performance of the *Ube3b*^−/−^ animals and occlude the behavioral readouts of Ube3b-associated neuronal deficit-linked changes. Despite these limitations, full *Ube3b* KO mice are a model of human systemic KOS, a metabolic disease strongly affecting the skeletal and nervous system [106].

## 20. Neuronal Roles of UBE3B

Our most recent work on the murine ortholog of UBE3B addresses specifically its cell-autonomous role in developing nervous system and involvement in formation of neuronal circuits. To circumvent the syndromic lethality and the overt non-neuronal phenotypes in the *Ube3b*^−/−^ mice, *Emx1*-driven, forebrain neuron and glia-specific mouse model was established, given a high expression of Ube3b in the cortex and the hippocampus. Ube3b associates with postsynaptic density fractions and regulates dendritic branching in a cell-autonomous manner. Conditional deletion of *Ube3b* in the forebrain is associated with decreased cortical volume, due to reduction of dendritic trees, and thinning of corpus callosum. Neurons and glia-restricted loss of *Ube3b* leads to increased spine density, a phenotype reversible by re-introducing Ube3b by in vivo transfection. Increased density of spines is in line with increased frequency of miniature excitatory postsynaptic potentials (mEPSPs). Regarding neuronal physiology, Ube3b maintains the ratio of NMDA to AMPA at the synapse and abundance of NR2A. Forebrain-restricted loss of *Ube3b* leads to altered activity of hippocampal circuit and induces complex behavioral changes. These include specific hippocampal circuitry-associated loss of spatial memory and increased social memory. Moreover, *Ube3b* cKO male mice exhibit a behavioral switch towards self-directed actions (like grooming and eating) at the expense of exteroceptive behaviors, such as exploration. On a molecular level, with regard to its synaptogenic function, Ube3b ubiquitinates γ-subunit of calcineurin, Ppp3cc and regulates its levels at the synapse [26]. Among the putative UBE3B interactors identified by affinity chromatography were regulatory subunits of protein phosphatases, as well [106]. Future research should determine how Ube3b regulates the activity of the phosphatases during neuronal development. On a biochemical level, Ube3b catalyzes formation of K48- as well as of K63-linked polyUb chains.

## 21. *Ube3b* Conditional and Conventional KO Mouse Models: Lessons Learned

The case of Ube3b and the abovementioned two reports reopens a discussion on the applicability of genetic mouse models and conclusions drawn based on neurodevelopmental phenotypes. While the *Ube3b*^−/−^ mouse model serves to study human KOS, it is not clear if the brain phenotypes described for the full KO animals are due to a loss of *Ube3b* or constitute secondary effects of systemic dyshomeostasis and/or other endophenotypes. The forebrain-specific *Ube3b* cKO does not serve as a comprehensive model of KOS, since it restricts the loss of the gene to excitatory neurons and glia, but offers a possibility to determine the cell autonomous roles of Ube3b in the cerebral cortex and the hippocampus. Notably, of the most significantly increased metabolites in *Ube3b*^−/−^ was urea, whose elevated levels are associated with brain inflammation, simplified dendritic arbors and reduced spine density in a mouse model of kidney failure [107]. Further research should determine the kidney function in *Ube3b*^−/−^ mice and address the kidney-brain axis regarding the systemic influence of uremic metabolites on the developing neurons. It is tempting to speculate that loss of spines and synapses in *Ube3b*^−/−^ mice is due to a non-cell autonomous mechanism. This should be verified in experiments addressing the effect of neuronal exposure to metabolites identified in plasma and cortex of *Ube3b*^−/−^ mice on spine density. Additionally, future experiments should address the effects of calcineurin and/or other protein phosphatase inhibition on synapse number both in *Ube3b* conditional and conventional mouse models. This is particularly thought-provoking regarding the UBE3A-driven regulation of PP2A activity and small molecule-mediated rescue of AS-like phenotypes in *Ube3a*^m−/p+^ mice.

According to the published works, Ube3b mRNA is depolarization-induced and calmodulin, a Ca^2+^-binding messenger, inhibits the enzymatic activity of Ube3b. On the other hand, Ube3b ubiquitinates and/ergo regulates the level of the γ-subunit of calcineurin, Ca^2+^-, and calmodulin-dependent serine/threonine protein phosphatase. Taken together, neuronal depolarization and Ca^2+^ are important regulators of Ube3b enzymatic activity in the developing neocortex, and in turn, the enzymatic activity of Ube3b controls the cellular responders to Ca^2+^ (Figure 4, Table 2).

## 22. Biological Functions and Clinical Importance of UBE3C: More Is Yet to Come

Of the three UBE3 ligases, neuronal roles of UBE3C remain uncharted. In the literature, UBE3C also exists as RTA-associated ubiquitin ligase (RAUL), or KIAA10. Since its identification through cDNA sequencing in myeloid cells [114], relatively little research concerning its role and function has been conducted. However, transcriptomic and proteomic approaches show that UBE3C is expressed throughout the body and enriched in skeletal muscles and the brain [115], where it interacts with human postsynaptic density [116] and such interaction undergoes dynamic scaling during sleep-wake cycle [117]. These findings are particularly relevant in the light of point mutations in *UBE3C*, resulting in single amino acid substitution in the catalytic domain region, described in ASD patients [39,40] (Table 1). Future studies should explore potential local UBE3C-mediated ubiquitination at the excitatory synapse. As is the case for UBE3A, UBE3C has been found in the nucleus [118]. Future research should explore a potential role of UBE3C in transcriptional control.

## 23. UBE3C in Human Disease

The involvement of UBE3C in various types of cancer has been most commonly reported. In many cases, upregulated UBE3C promotes the progression of the disease by tumorigenic and metastatic effects, associated with an increase in cellular migration and proliferation [110,119,120,121,122]. Additional populational genetic studies have associated *UBE3C* with disorders such as asthma [123], stroke [124], major depressive disorder [125,126], cocaine addiction [125], Parkinson’s [127], and Down’s syndrome [128].

## 24. UBE3C as a Proteasomal Regulator

UBE3C seems to have a distinct biological role as an E3 ligase that directly influences the cellular proteolytic machinery. Firstly, UBE3C and a DUB Usp14 continuously cycle on and off 26S proteasomes in ubiquitinated substrates-dependent manner and regulate cellular proteostasis [129]. UBE3C has also been described as a positive regulator of proteasomal processivity by propelling efficient proteolysis of accumulating partially degraded protein fragments [130]. Additionally, UBE3C-mediated ubiquitination of a proteasome subunit Rpn13 constitutes an autoinhibitory mechanism to prevent the attachment of Ub conjugates with stalled proteasomes [112]. Interaction with proteasomes and regulation of its physiology seems to be a common feature of UBE3A and UBE3C. Among other canonical degradation substrates of UBE3C are mostly transcriptional regulators (Table 2).

## 25. UBE3C Generates K29-Linked Ub Chains

Aside from its well-established role in proteolysis involving its inherent ability to conjugate K48-linked polyUb [131], UBE3C forms non-canonical K11- and K29-linked chains [27]. The role of these modification types has been poorly understood. It seems that K29-linked Ubs are found in heterotypical branched chains [28], implicating UBE3C as an important regulator of Ub chain morphology. Research on non-canonical Ub linkages is hampered by sparsity of available tools, however, K29 ubiquitination serves as a negative regulator of Wnt/β-Catenin [132], mediates viral infection, innate antiviral response [133,134], and is required for ER-mediated degradation. All of these processes are also associated with the function of UBE3C [113,122,135]. Considering a rather strong evidence of UBE3C involvement in the pathology of several human neurological diseases, future research should elucidate the cellular and molecular outcomes of UBE3C enzymatic activity and K29 ubiquitination in developing cortex.

## 26. Novel Concepts and Concluding Remarks

Despite classical grouping into one family, UBE3 ligases display fundamentally different characteristics, supported by the phylogenetic, biochemical, physiological, and clinical evidence. Each of UBE3 constitutes a separate biochemical entity, with specific ubiquitination pathways that they supervise, necessary for neuronal development. Based on the recent evidence, UBE3A has been shown to operate in the nucleus in an isoform-specific manner, where it globally regulates neuronal homeostasis. Other isoforms of UBE3A seem to have opposing effects in terms of their morphoregulatory and synaptogenic potential, and might operate locally in neurons, some of them not necessarily as enzymes. On top of this, UBE3A seems to modulate cellular proteolytic machinery. UBE3B associates with synaptic fractions, possibly forming a bigger protein complex, involving calcineurin. It would be interesting to test if its metabolic role is achieved in a global manner, by operating in different cellular compartments, similarly to UBE3A. Proteasome function depends on the association with ASD-linked UBE3C, able to assemble non-canonical polyUb chains. It would be extremely important to address developmental roles of not solely the ligase, but also of proteasome, its activity and non-degradative ubiquitination in ASD mouse models.

Future research will bring answers not only in terms of detailed molecular context of their action, but also regarding the biochemistry of non-conventional ubiquitination and its cellular and physiological consequences. This will aid clinical trial design for neurological and neurodegenerative diseases and cancer and shed light onto the complex machinery at play in the development and homeostasis of neuronal circuits.

## Figures and Tables

**Figure 1 cells-09-02455-f001:**
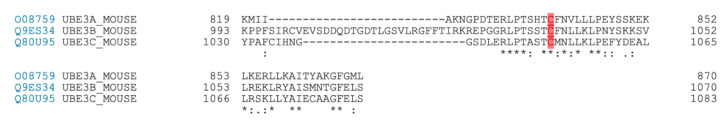
Sequence alignment of the C-termini of Ube3 ligases (Uniprot accession numbers on the left). Numbers on the left and right indicate the extreme amino acid positions. Highlighted in red is the active site catalytic cysteine. Symbols “*”, “:”, “.”, denote the degree of single residue homology.

**Figure 2 cells-09-02455-f002:**
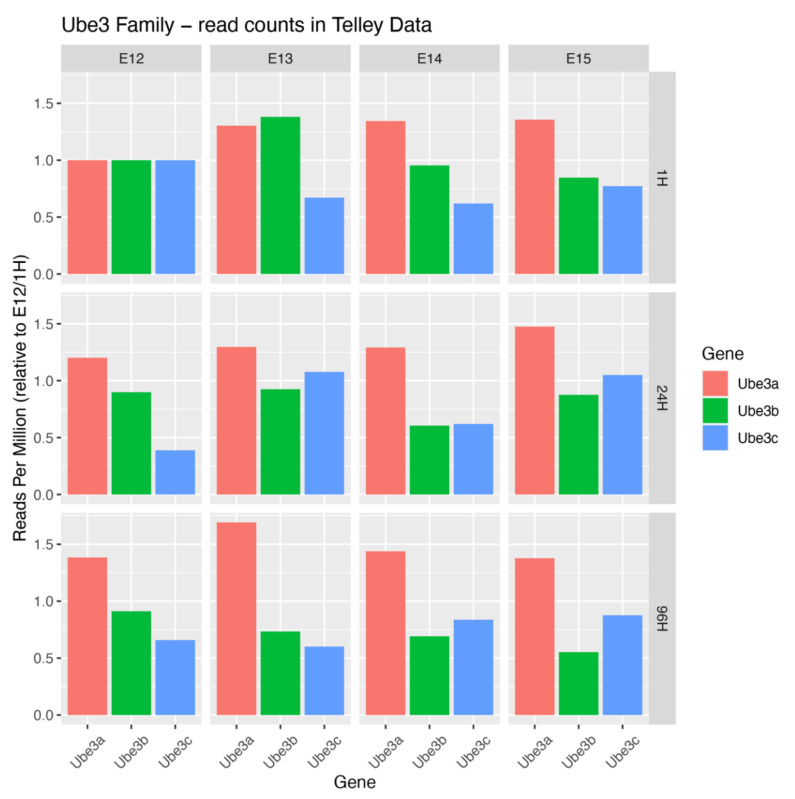
Expression levels of Ube3 ligase genes in developing cortex. To plot expression levels of Ube genes, we downloaded the digital gene expression matrix (DGE) provided by Telley et al. [3], in which cells are annotated by the time of collection, and time since FlashTag labelling. We grouped all cells with the same annotation to counts per-gene, and by summing all such values we calculated a “library size” combination of variables, normalizing the counts to control for the number of cells of each type collected. For each gene, we then normalized the count values to the value at the earliest time of collection and “flashtagging”, in order to plot relative change in expression over time.

**Figure 3 cells-09-02455-f003:**
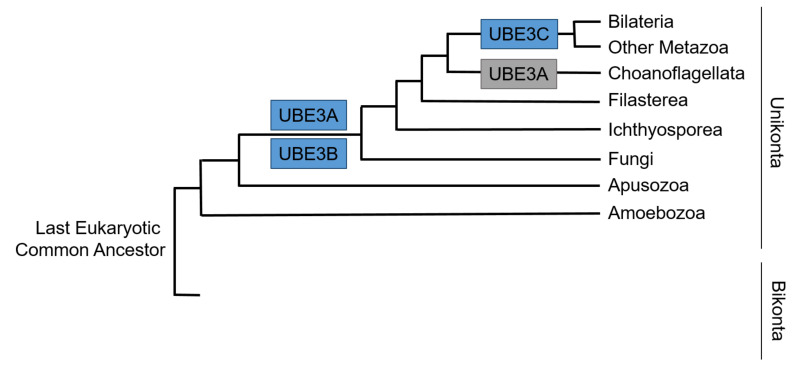
Simplified pattern of acquisition of Ubiquitin Ligases E3s (UBE3s) across the tree of life of *Eucaryota* based on [44]. Gains are colored in blue and losses in grey.

**Figure 4 cells-09-02455-f004:**
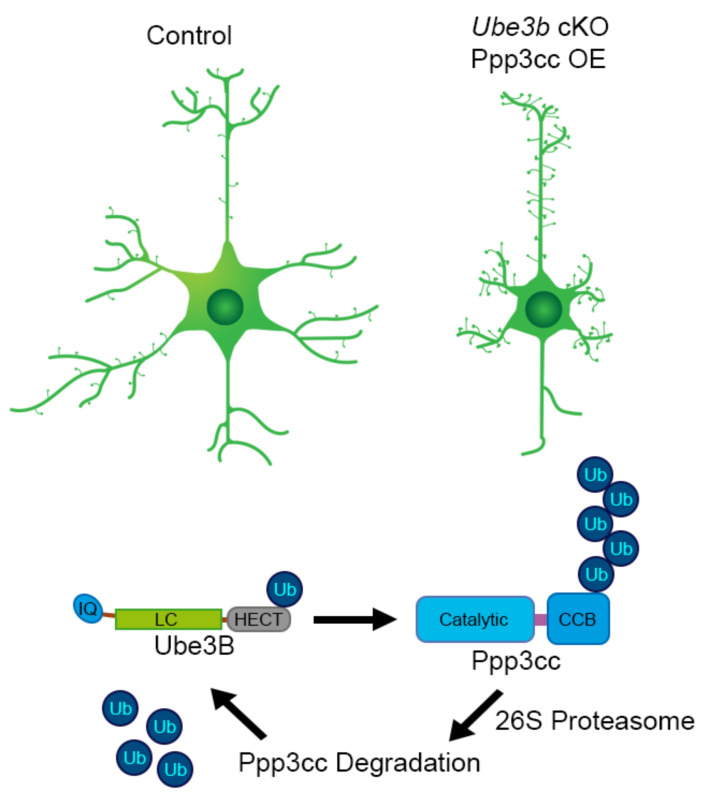
Ube3b-mediated Ppp3cc regulation. In the forebrain-specific *Ube3b* cKO mouse, cortical and hippocampal neurons exhibit simpler, less arborized dendritic trees. This phenotype is mimicked, once neurons overexpress gamma-subunit of calcineurin, Ppp3cc, ubiquitination substrate of Ube3b. LC, low complexity; CBB, calcineurin B and calmodulin binding. PolyUb chain on position on Ppp3cc is randomly placed on the scheme to depict Ube3b-mediated ubiquitination.

**Table 1 cells-09-02455-t001:** Amino acid residues necessary for UBE3C enzymatic activity.

Mutation	Region	Autoubiquitination Activity	Reference
K903R	HECT domain	inactive	[38]
Q961A, Q961E	HECT domain	reduced	[38]
S1049H	HECT domain	reduced	[38]
S845F	HECT domain	under investigation	[39]
F996C	HECT domain	under investigation	[40]

**Table 2 cells-09-02455-t002:** Ubiquitination substrates of Ube3b and Ube3c.

Ube3 Ligase	Substrate	Ub Chain	Reference
Ube3b	BCKDK	pan-Ub	[106]
	Ppp3cc	pan-Ub, K48	[26]
Ube3c (KIAA10, RAUL, yeast Hul5)	TIP120B	pan-Ub	[108]
	AHNAK-p53	pan-Ub	[109]
	Annexin A7	pan-Ub	[110]
	RPN13	pan-Ub	[111,112]
	IRF3/IRF7	pan-Ub	[113]

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
