# Peer review of "Molecular Evolution, Neurodevelopmental Roles and Clinical Significance of HECT-Type UBE3 E3 Ubiquitin Ligases"

_cells, 2020, doi:10.3390/cells9112455_

Round 1
Reviewer 1 Report
This review article focuses on a class of E3 ligases that mediate the ubiquitination of target substrates in eukaryotic cells. While the role of ubiquitination has been established both as a destruction tag and a mediator of cell signalling, the precis function of specific E3 ligases remains relatively unexplored. This is due to several challenges most notably the sheer number approx. of E3 ligases encoded by the human genome as well as difficulties in identifying their biological substrates. In this review articles the authors focus on the UBE3 family of ligases. The UBE3 family of HECT ligases, comprising UBE3A, UBE3B and UBE3C are predominantly expressed in neuronal cells; however, their exact functions and the pathways they regulate are poorly defined. This review provides a comprehensive survey of the literature on the structure and role of the UBE3 family members in human disease and in knockout mouse models.
Strengths:
- Well-structured review article with a clear focus on what is known and unknown. The authors place the findings in context and do suggests similarities with other known signalling pathways.
- Good use of figures showing UBE3 expression levels in developing cells.
Overall relevant figures and tables that support the content of the article. - The fact that UBE3 members are associated with human diseases such as Angelman's and Kaufman's syndrome is of particular interest and may suggest strategies for therapeutic intervention.
- The authors do give focus in UBE3C which is one of the least described family members and potentially of interest for the development of anti-cancer strategies. UBE3C shuttles on and off the proteasome (along with a DUB USP14 that has been suggested as a target in cancer and neurodegenerative diseases). The authors do a good job of highlighting the available evidence and placing UBE3C in the context of other known signaling pathways. I feel that this is definitely the most interesting aspect of the article and suggests potential of UBE3C as a target of proteasome inhibitor strategies for cancer.
Weaknesses:
- No major weaknesses (maybe slight language edit to improve readability) but nothing that would hinder publication.
Overall a solid and deeply researched review article of general interest to the field.
Reviewer 2 Report
The manuscript “Molecular evolution, neurodevelopmental roles and clinical significance of HECT-Type UBE3 E3 Ubiquitin Ligases” from Ambrozkiewicz et al. reviews the knowledge about UBE3 ligases, in particular in relation to their role in the brain and their interconnections.
The idea is appealing and of great interest. However, the focus, the clarity, and the explanation of the topics in this review are quite unfocused and not exhaustive. The article presents several weaknesses and in several points is not clear (sometimes even not precise).
All the manuscript is quite confusing, several parts have not enough references while other parts (e.g. the part on UBE3A and AS) presents old conclusions that are today redirected by the most recent publications in the field. The conclusions of this manuscript don’t report about new ideas/concepts either not a good resume + explanation of the present knowledge. This review needs revision and rewrite work.
Briefly, I reassume some major issues:
line 44-85: The text does not explain the general topic of neuronal networking and brain formation, where E3 ligases act. It is confusing, if not wrong (line 55-radial glial cells DO NOT differentiate into neurons!).
Line 149 Angelman is due to the lack of UBE3A! The mutation is one of the genetic mechanism, that accounts for 10% of AS.
line 160-247: in this part the authors do not introduce the general basis of the topic and why this part can be of interest
Line 314- Ube3a is expressed biallelically in glia so at higher level than in neurons.
In general the manuscript does not take in account the pathological status due to the increase of UBE3A levels (15Dup autism and the other ASD due to duplication/triplication of Ube3a).
Abstract, line 25: E3 are 60-70, not 600-700
Line 340-41: no ref is provided. Ube3a is for sure implicated in neuronal morphogenesis, but its role is not yet clear, there are several contrasting reports: sometime UBE3a loss impacts neuronal morpho-functional features, sometimes not, it depends where, when in the brain and sometimes also which isoform is considered (e.g. Valluy et al Nat Neurosci 2015).
Round 2
Reviewer 2 Report
The authors increased the scope and the clarity of their conclusions, as needed. The introduction of shorter paragraphs with explicative titles, and some sentences for introducing the authors’ point of view have been useful to clarify the manuscript.
Some comments:
- The judgment of language at the level of 1/5 was wrong, by the reviewer’s mistake. The language is more than ok and the correct judgment was 5/5, sorry.
- Line 314- Ube3a is expressed biallelically in glia so at higher level than in neurons. The authors are right, my comment was intended for the pathological (AS) conditions.
- Line 372- Isoform-specific neuromorphoregulatory roles of UBE3A
Here the authors well introduced a new paragraph but there is a bit of confusion about the UBE3A isoform nomenclature, which is different between human and mouse. As resumed in ref 69, in humans, there are three known UBE3A protein isoforms. Human isoforms 2 and 3 only differ from human isoform 1 by 23 and 20 amino acids, respectively, at their N termini. The mouse only has two protein isoforms of UBE3A that retain ubiquitin ligase activity, with the shorter of these two (termed isoform 3 in mouse) being equivalent to human isoform 1 and the longer (termed isoform 2 in mouse) being most similar to human isoform 3. Mouse isoform 1 is predicted to lack ubiquitin ligase activity (it seems to acts as RNA, as in ref70), and an analogous isoform does not exist in humans, while an analogous form of human isoform 2 does not exist in the mouse.
Therefore it should be added at line 382 that ref 69 refers to humans, so human isoform 1. While for ref 70, the work refers to murine isoforms and neurons. I suggest adding murine or human to the isoforms to better clarify.
- line 401: referring to ref 47, the authors should mention also that the UBE3A effect on ARC has been later reported as indirect, accordingly to https://doi.org/10.1073/pnas.1302792110.